



# Using a deep neural network to detect methane point sources and quantify emissions from PRISMA hyperspectral satellite images

Peter Joyce[1,2,3], Cristina Ruiz Villena[4,5], Yahui Huang[2,3], Alex Webb[4,5], Manuel Gloor[1], Fabien H. Wagner[6,7], Martyn P. Chipperfield[2,3], Rocío Barrio Guilló[4], Chris Wilson[2,3], and Hartmut Boesch[4,5]

[1]*School of Geography, University of Leeds, Leeds, United Kingdom*

[2]*National Centre for Earth Observation, University of Leeds, Leeds, United Kingdom*

[3]*School of Earth and Environment, University of Leeds, Leeds, United Kingdom*

[4]*University of Leicester, Leicester, United Kingdom*

[5]*National Centre for Earth Observation, University of Leicester, Leicester, United Kingdom*

[6]*Institute of Environment and Sustainability, University of California, Los Angeles, CA, USA*

[7]*Jet Propulsion Laboratory, California Institute of Technology, 4800 Oak Grove, Pasadena, CA 91109, USA*

*Correspondence to*: Hartmut Boesch (hb100@leicester.ac.uk)

**Abstract.** Anthropogenic emissions of methane ($CH_4$) make up a considerable contribution towards the Earth's radiative budget since pre-industrial times. This is because large amounts of methane are emitted from human activities and the global warming potential of methane is high. The majority of anthropogenic fossil methane emissions to the atmosphere originate from a large number of small (point) sources. Thus, detection and accurate, rapid quantification of such emissions is vital to enable the reduction of emissions to help mitigate future climate change. There exist a number of instruments on satellites that measure radiation at methane-absorbing wavelengths, which have sufficiently high spatial resolution that can be used for detecting highly spatially localised methane 'point sources' (areas on the order of $km^2$). Searching for methane plumes in





methane sensitive satellite images using classical methods, such as thresholding and clustering, can be useful but are time-
consuming and often inaccurate. Here, we develop a deep neural network to identify and quantify methane point source
emissions from hyperspectral imagery from the PRecursore IperSpettrale della Missione Applicativa (PRISMA) satellite with
30-m spatial resolution. The moderately high spectral and spatial resolution as well as considerable global coverage and free
access to data make PRISMA a good candidate for methane plume detection. The neural network was trained with simulated
synthetic methane plumes generated with the Large Eddy Simulation extension of the Weather Research and Forecasting
model (WRF-LES), which we embedded into PRISMA images. The deep neural network was successful at locating plumes
with F1-score, precision and recall of 0.95, 0.96 and 0.92, respectively, and was able to quantify emission rates with a mean
error of 24%. The neural network was furthermore able to locate several plumes in real-world images. We have thus
demonstrated that our method can be effective in locating and quantifying methane point source emissions in near real time
from 30-m resolution satellite data which can aid us in mitigating future climate change.

## 1 Introduction

Methane ($CH_4$) is a powerful greenhouse gas with a warming potential which per unit mass emitted is 84 times larger than
for carbon dioxide over a 20-year period (Stocker et al., 2013). Emissions of methane as a result of human activities have
contributed one quarter of climate warming since preindustrial times (Etminan et al., 2016). A large proportion of
anthropogenic methane from industrial sources originates from point sources such as coal mines and oil and gas production
facilities (Saunois et al., 2020). Furthermore, these emissions are generally underestimated by inventory-based approaches
(Alvarez et al., 2018; Karion et al., 2013; Zavala-Araiza et al., 2015). A large proportion of these anthropogenic emissions
originates from a small number of strong point sources due to oil and gas production equipment malfunction (Brandt et al.,
2016; Duren et al., 2019; Zavala-Araiza et al., 2017). Consequently, much of the methane emitted from such sources could
be reduced at no net cost (IEA, 2017; Ocko et al., 2021). Acting to reduce methane emissions in this sector can be one of the
most cost-effective methods of mitigating against further climate change.

Methane point sources from oil and gas production are typically small in extent and emissions difficult to quantify and variable
in time (Allen et al., 2013; Frankenberg et al., 2016). The primary challenge faced when estimating methane emissions from
point sources from satellite data comes from the relatively low spatial resolution (in the order of kilometres) of satellite
imagery from dedicated sensors such as the Greenhouse Gases Observing SATellite (GOSAT) (Kuze et al., 2009) and the
TROPOspheric Monitoring Instrument (TROPOMI) (Levelt et al., 2006). These sensors typically have high spectral
resolution of methane absorption bands in the shortwave infrared (SWIR) range of the electromagnetic spectrum to provide
accurate measurements with high precisions of around 10-20 parts per billion (ppb) (Lorente et al., 2021; Parker et al., 2020).
SWIR bands can also be effectively utilised to detect and quantify point sources from lower spectral-resolution sensors (Jacob
et al., 2016; Duren et al., 2019). Recent hyperspectral spaceborne imaging spectrometers contain hundreds of spectral



channels in the visible-shortwave-infrared range with spectral resolution typically around 10 nm and spatial resolutions of
tens of m. Due to their spatial and spectral resolution, they have been identified as useful new tools for identifying and
quantifying methane point source emissions. PRecursore IperSpettrale della Missione Applicativa (PRISMA), developed and
operated by the Italian Space Agency (ISA) since 2019, is the first hyperspectral mission where the satellite imagery has been
openly released to the scientific community. The satellite consists of a panchromatic camera and an advanced hyperspectral
instrument that measures radiances in approximately 250 bands between 400 and 2500 nm. The instrument has a spatial
resolution of 30 m, a swath of 30 km, and a 12-nm spectral resolution (Galeazzi et al., 2008). How to best extract information
on the location and extent of methane plumes is not yet fully established. Successful detection of methane point sources from
PRISMA using a matched-filter retrieval technique has been reported by Guanter et al. (2021), albeit with a strong dependence
of detection accuracy on surface type. In particular, brightness and homogeneity of the satellite images were identified to
significantly influence the accuracy of methane detection techniques.

Current approaches for detecting methane point sources and quantifying emission rates are time-intensive, laborious, and
prone to errors owing to the substantial human intervention required. They typically involve a spectral analysis to infer
methane column mean mixing ratios (Thorpe et al., 2014) followed by a methane plume detection method (often based on
thresholding and clustering) and finally the integrated mass enhancement (IME) method to estimate the emission (Varon et
al., 2018). Previous efforts utilising spaceborne imaging spectrometers to quantify methane point source emission rates have
proved successful, but often with large errors of source detection and emissions estimates. The IME method yielded errors
between 5-12% using 50-m resolution Greenhouse Gas Satellite - Demonstrator (GHGSat-D) imagery (Varon et al., 2018).
However, this uncertainty estimate does not include errors from unknown wind speed and direction, which are both highly
uncertain, thus uncertainties are effectively much larger. The multi-band multi-pass (MBMP) method was successful in
quantifying methane point source emissions from Sentinel-2 multispectral instrument (MSI) imagery with precision between
30% and 90% (Varon et al., 2021). The primary limitation of this approach is surface interference (Cusworth et al., 2019)
which leads to artefacts and false anomalies, which can be mistakenly attributed to emission plumes. This is a major
disadvantage for multi and hyperspectral missions because the better the resolution (and the greater the number of channels),
the better the discrimination between the surface and methane absorption. Thus, producing a model that minimises such errors
and can automatically locate methane sources would make emission monitoring from space faster, more reliable, and more
scalable, thus providing an invaluable tool to aid mitigation. A first effort has also been made to estimate emission rates from
AVIRIS-NG data using a neural network and without utilising wind speed and direction data. These estimates were subject
to an error of roughly 30% of the emission rates (Jongaramrungruang et al., 2019). It is apparent that the noise in the satellite
data, the lack of accurate wind data, and the complex structures of methane plumes make it difficult to model emission rates
accurately via traditional approaches.



In recent years, deep neural network methods have improved rapidly. LeNet (Lecun et al., 1989) was one of the earliest
convolutional neural networks (CNNs) and was used successfully to identify handwritten digits. This work laid the
foundations for using artificial intelligence to obtain meaningful information from image data (known as *computer vision*).
Deep learning models entered the mainstream following considerable reductions in model training time through the utilisation
of graphics processing units (GPUs) (Oh and Jung, 2004). Deep learning was then revolutionised for image classification
with the introduction of AlexNet (Krizhevsky et al., 2012). CNNs have since been applied to self-driving cars (e.g., Nugraha
and Su, 2017), discovering new drug treatments (e.g. Wallach et al., 2015), facial recognition (e.g. Matsugu et al., 2003), and
many other applications. The ease with which deep neural networks can be trained and deployed has also improved
considerably in recent years, partially due to the development of application programming interfaces (APIs) such as Keras
(Chollet, 2015). This has been supplemented by the increasing ubiquity and decreasing costs of GPUs and cloud computing
servers, which together have enabled deep learning models to be trained rapidly and at a relatively low cost. Currently, work
utilising deep neural networks has already proven to be considerably more effective than classical methods to detect point
source emissions of nitrogen dioxide ($NO_2$) (Finch et al., 2021).

More recently, a deep neural network has been used to quantify methane point source emissions using the airborne AVIRIS-
NG instrument (Jongaramrungruang et al., 2022). In this study, a CNN was trained on synthetic plumes inserted into real
images to extract features present in plumes of varying intensities and with differing wind speeds to locate and quantify the
emission rates of the point sources. Jongaramrungruang et al. (2022) estimated emission rates of plumes with a mean absolute
error of 17% for emissions larger than 40 kg hr$^{-1}$. The classification accuracy (determining whether a plume is present in an
image) was 90% when testing plumes with emission rates above 100 kg hr$^{-1}$, however, the accuracy dropped to 50% for
emission rates around 50-60 kg hr$^{-1}$. The spatial and spectral resolution of the aircraft data used in this study (AVIRIS-NG)
has far higher spatial and spectral resolution than PRISMA, thus making methane detection prone to lower errors. However,
PRISMA data is publicly available and covers a far larger spatial range with regular repeat measurements, thus making it a
superior resource for rapid detection of methane point source emissions across many regions on earth. Thus, a deep neural
network that is capable of utilising PRISMA data to detect methane emissions could be very effective in our efforts to mitigate
future climate change.

In this study, we produced pseudo-observations of simulated synthetic methane plumes generated with the Large Eddy
Simulation extension of the Weather Research and Forecasting model (WRF-LES). These simulated plumes were then
embedded into an array of PRISMA images and used as training data for a novel neural network architecture that aimed to
produce masks of the locations of methane plumes and estimate their emission rates from PRISMA satellite imagery. The
effectiveness of this model was then tested on images of real-world plumes. The techniques utilised here can be adapted to
locate and quantify emission rates using any satellite imagery with suitable shortwave-infrared bands, or applied to detecting
other greenhouse gases, such as carbon dioxide ($CO_2$).



## 2 Methods

### 2.1 Simulating methane plumes with WRF-LES

The Weather Research and Forecasting (WRF) model system has comprehensive and multiple capabilities for studying atmospheric phenomena from global down to large eddy scales. The default large eddy simulation case (LES) of the WRF V4.2.2 was used and modified to simulate methane plumes for a single point source with a releasing rate of 1000 kg hr$^{-1}$. The default LES case does not consider clouds, radiation, or topography, but includes surface physics and 1.5-order TKE (Turbulent Kinetic Energy) prediction scheme (WRF model User's Guide: https://www2.mmm.ucar.edu/wrf/users/). A constant thermal flux of 100 W m$^{-2}$ was applied at the surface to drive the turbulence. Two nested domains with one-way nesting were deployed in the simulations. The outer domain had a size of 5.4 km x 6.3 km with 90 m horizontal resolution and periodic boundary conditions. The inner domain had a size of 3.6 km x 4.5 km with 30 m horizontal grid spacing and 30 m vertical resolution, and flow-dependent boundary conditions for scalars. The plume was only released in the inner domain after a 3-hour spin-up run. The total running time is 5 hours, and the final 2-hour run was considered for the training, test, and validation data.

We designed 15 scenarios consisting of 5 different southerly wind speeds ranging from 1 m s$^{-1}$ to 9 m s$^{-1}$, each of which was uniformly applied from the surface to the model top, and 3 different patterns of potential temperature vertical profiles (Figure S1). The potential temperature in the scenarios is specified as 290 K from the surface to one of the 3 different mixing depths of 500 m, 800 m, and 1100 m (Figure S2). Above the mixing depth, there is an inversion layer of 700 m with a vertical gradient of potential temperature of 0.009 K m$^{-1}$ applied from the top of the mixing layer to the model top. For each simulation, the CH$_4$ distribution is saved once every minute and thus there are 120 different scenes for a two hour simulation. Altogether there are 1800 scenes for the 15 simulations in the data, where the plume was integrated over vertical columns. Figure 1 shows one snapshot of a plume with initial conditions of 3 m s$^{-1}$ southerly wind and 800 m mixing depth 30 minutes after release.



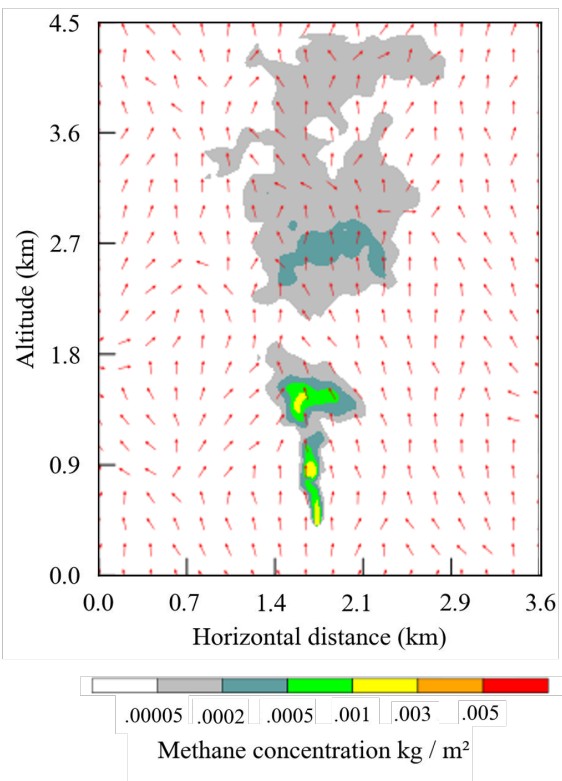

Methane concentration kg / m²

144

**Figure 1: Snapshot of a simulated plume 30 minutes after release for initial conditions of 3 m s⁻¹ southerly wind and 800 m mixing depths. Red arrows indicate wind direction at the moment of the snapshot.**

**2.2 Satellite data retrieval**

Methane absorbs solar radiation at a set of shortwave-infrared wavelengths that are well known and documented in spectroscopic databases. The absorption of light by methane in the atmosphere therefore alters the reflected sunlight measured by the satellite in a very predictable way that allows us to quantify the amount of methane along the light path. Here we use a data-driven retrieval algorithm to estimate the methane enhancements from reflected sunlight using statistical methods based on the work by Thorpe et al. (2014). This type of simple and fast retrieval method is commonly used for instruments with comparably low spectral resolutions, for which a more sophisticated, so-called full-physics approach provides no extra benefit.

The relationship between the spectral intensity at each point in the satellite spectra and the column enhancement of methane in the scene is represented by a methane Jacobian vector, which describes the change in the logarithm of the intensity $I_k$ in band $k$ with respect to the   column enhancement of methane $C_{CH4}$. The spectral variation of the background of the scene (i.e. outside of the plume) is approximated by a number of Principal Components of all measured spectra combined derived using



the Principal Component Analysis (PCA) method. We perform the PCA on the logarithm of measured spectra of the scene
and select the singular vectors (principal components) that best describe the spectral variability of the scene. The optimal
number of singular vectors was determined by trial and error, and was found to be the first three. We then concatenate these
vectors with the methane Jacobian to construct the matrix **J** with dimension 4x number of PRISMA bands, which we use
along with the logarithm of the measured radiances, **y,** to find a vector **W** that minimises the cost function in a linear least
squares fit for each pixel:
$\|y - JW\|^2$ ,                                                                                         (1)

The modelled radiance **F** is calculated from **J** and **W** as follows:
$F = JW$ ,                                                                                               (2)

We can then rewrite Eq. (2) as the sum of the background ($k$) and CH4 ($c+1$) components of the radiance:
$F(W,J) = \sum_{k=1}^{c} J_k \cdot W_k + J_{c+1} \cdot W_{c+1}$ ,                                         (3)

where $c$ is the number of singular vectors used. Thus, the modelled logarithmic radiance F(W, J) is a linear combination of
the singular vectors, $J_k$, the CH$_4$ Jacobian, $J_{c+1}$, and their weights, $W_k$ and $W_{c+1}$, respectively. This method is described in
more detail in Thorpe et al. (2014). Since the wavelengths scale for each across-track pixel of a PRISMA image are different,
it is necessary to infer the Principal Components for each column in the across-track direction separately.
**2.3 Training data generation**
We generated synthetic datasets to train the machine-learning model by combining PRISMA images with the synthetic plumes
simulated with WRF-LES (described in section 2.1). We use the SWIR spectral radiance from PRISMA Level-1b data as
well as the RGB bands. These datasets come with pixel quality and cloud mask information, which we apply in our data
preparation process. We selected 36 different PRISMA background images to cover a wide range of scenes representative of
places where methane plumes might be expected (Table S1). These images also cover a range of different dates throughout
the ~3 years of PRISMA data available in the archive, to account for different illumination conditions. All the selected scenes
have less than 1% cloud cover, and any pixels flagged as cloudy in the PRISMA product were excluded from the analysis.

A total of 9700 image tiles were generated for training, each tile with a size of 256 x 256 pixels. The tile size was deliberately
selected as a power of two to optimise the model performance. Each tile was selected at random from one of the 36
1000x1000-pixel PRISMA background scenes, and a synthetic methane plume subsequently embedded in it. The synthetic
plume was also selected randomly from the WRF-LES simulations, with the following parameters also randomised following
a uniform distribution:




193 -  **Time step**: between 1 and 120 seconds (Figure S3).

194 -  **Plume origin**: any point within the background scene tile, excluding the areas near the edges to avoid missing parts

195   of the plume.

196 -  **Emission rate**: all simulated plumes have a 1000 kg hr$^{-1}$ emission rate, so we applied a scaling factor between 0.1

197   and 10 to have a range of emissions between 100 and 10,000 kg hr$^{-1}$ (Figure S4).

198 The synthetic plumes from WRF-LES are first converted into maps of methane vertical column densities in molecules cm$^{-2}$.

199 The original plume simulations are all carried out for an emission of 1000 kg hr$^{-1}$ and the scenarios for different emission

200 rates are obtained by scaling the simulated concentrations. Each plume is inserted into the background PRISMA image tile

201 by modifying the PRISMA SWIR radiances according to the Beer-Lambert law for absorption. Methane columns are

202 converted into optical depth for each band using a representative methane absorption cross-section for each band computed

203 from the HITRAN database (Gordon et al., 2022) for a temperature of 293K and pressure of 1 atmosphere. Each of the 9700

204 training datasets contain: 38 PRISMA radiance bands (3 RGB, and 35 SWIR (2100 - 2365 nm) channels) and the synthetic

205 plume (i.e., the "true" methane enhancements to be used as labels in the model).

206 **2.4 Training data processing**

207 Each PRISMA sub-image (256 x 256-pixel tile) was normalised by subtracting the mean and dividing by the standard

208 deviation (std) of the whole collection of training images such that the mean of all the images was 0 and the std was 1 for

209 each band. This data normalisation step is standard when using deep neural networks as it is understood to optimise the

210 training time. Following on from this, the undefined (NaN) values present in the images were changed to equal the mean

211 value of each band in the respective image. These NaN values correspond to either invalid (e.g., saturated) or cloudy pixels.

212

213 Every time an image was retrieved during the training process, data augmentations were randomly applied. The augmentations

214 were as follows: rotation by a multiple of 90˚, and horizontal and vertical flipping. No brightness and contrast augmentations

215 were made because the quantification of methane plumes relies on the specific band information inside the plume region. The

216 purpose of data augmentation was to increase the data volume, to reduce overfitting, and improve the ability of the model to

217 produce accurate results with data that is different to the training data.

218

219 To predict the methane concentration, it was first necessary to model the methane plume mask (binary classification of

220 plume/non-plume) because the vast majority of pixels in the training images did not contain a plume (zero-inflated data). An

221 initial methane concentration threshold of $8 \times 10^{18}$ molecules cm$^{-2}$ was chosen as it was the cut-off point where the plumes

222 were no longer visible. Furthermore, training the model with a lower threshold resulted in non-convergence. After the model

223 was trained at the $8 \times 10^{18}$ molecules cm$^{-2}$ threshold, it was possible to continue training the model at a lower threshold. Thus,




we tested training the model at $5\times10^{17}$ molecules cm$^{-2}$ increments until the validation loss dropped substantially. The lowest
threshold where this was the case was $4\times10^{18}$ molecules cm$^{-2}$. This final step is important because it increases the range for
which the model can locate and quantify methane emissions.

**2.5 Deep neural network architecture and training process**

The training of the neural network was split into 4 steps. First, the model was trained to locate the regions of the image
containing a plume via binary semantic segmentation. Next, the column enhancements of methane were predicted inside the
region of the estimated plume mask from the first stage. Following on from this, the emission rate of the plume in the image
was estimated. To ensure that the emission rate estimates would equal zero when no plume was present, an intermediate
prediction layer was included where a binary classification was made regarding whether a plume was present in the image or
not. At each stage of the model, the input was a concatenation of the input satellite image and all the previous outputs (Figure
2). To optimise the training of the model weights, each portion of the model was trained alone such that the weights in all the
other parts were not being updated. The parts of the model were trained in order moving downwards across the models
depicted in Figure 2. The loss function to predict the plume mask was as follows:
$\text{Loss}_{\text{mask}} = 1 + BC - SDC$ ,                              (4)
Where $BC$ is binary cross entropy, $SDC$ is the Sørensen-dice coefficient defined as follows:
$SDC = \frac{2TP}{2TP + FP + FN}$,                              (5)
where TP is true positive, FN is false negative, and FP is false positive. This loss function was chosen because of the large
number of non-plume pixels present in the image. The loss function for the mask concentration was mean squared error
(MSE), a standard choice for regression modelling. For the binary classification part of the model, binary cross-entropy was
chosen, which is common for solving 1-dimensional binary problems. Finally, for the emission rate part of the model, MSE
was chosen as the loss function until the validation error started to plateau, after which, the model was only trained on images
containing plumes and mean absolute percentage error was given as the loss function. This was done to ensure that the
proportion error was minimised rather than the absolute error. Mean absolute percentage error was not used throughout the
whole training process because it was important that the model was trained on some images with no plumes (so an emission
rate of zero could be possible) and mean absolute percentage error produced very high loss values when false positives were
made by the model.
The two encoder CNNs have identical architectures except the activation function at the end of the binary classification model
has sigmoid activation because the predictions are constrained between 0 and 1, and the emission rate estimator has a ReLU
activation function.





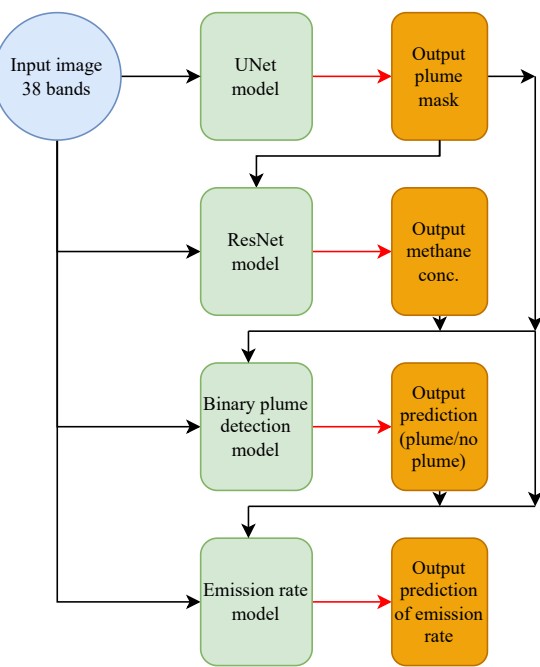


**Figure 2: Structure of the neural networks used in this study. Green boxes indicate portions of the neural network, orange boxes**
**indicate predictions made by each stage of the neural network. Black lines indicate flow of data into models, and red lines indicate**
**predictions resulting from a model.**
**2.5.1 Estimating plume masks**
Estimating the mask of a methane plume involved using a similar architecture to a UNet model (Ronneberger et al., 2015)
(Figure 3). UNet models consist of two paths; the first is the encoder, which captures the context in the image and is composed
of convolutional and max pooling layers. The second path is the decoder, which enables localisation of the features captured
by the encoder and consists of convolutional and upsampling layers (Ronneberger et al., 2015). In our model architecture,
there is an additional $1\times1$ convolutional layer with 64 filters at the beginning because methane plumes are associated with
anomalies in certain SWIR bands of the PRISMA imagery. Methane is not absorbed in the visible bands; thus, their inclusion
helps the neural network to distinguish between plume and non-plume by providing information on the background of the
image. Methane plumes can be identified based on the typical spatial structures that form as a result of turbulence and
advection in the atmosphere, as well as the variations in methane-absorbing bands compared with the background landscape.
It is the latter reason why an additional $1\times1$ convolutional layer was deemed to be helpful in improving the accuracy of the
model.



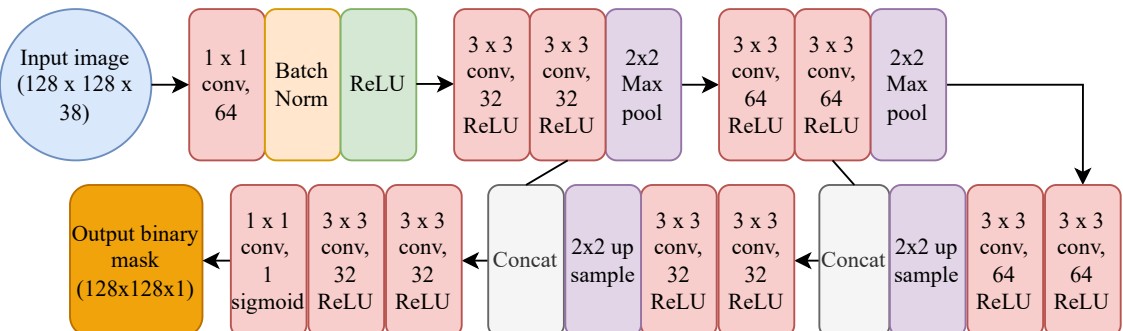


**Figure 3: Architecture of the deep neural network for the UNet portion of the model. 1 × 1 conv, 64 refers to a convolutional filter with kernel size 1 × 1 and 64 filters. Batch Norm refers to a batch normalisation layer, Concat refers to a concatenation between the inputs to that layer, 2 x 2 Max pool refers to a max pooling layer with pool size 2, and 2 x 2 up sample refers to upsampling layer with size 2. ReLU and sigmoid refer to the Rectified Linear Unit and sigmoid activation functions respectively.**

### 2.5.2 Estimating methane column enhancements inside plumes

Estimating the methane column enhancement within the plumes predicted in section 2.4.1 uses a concatenation of the input image and the mask predictions. This step to aid the estimation of methane concentrations is necessary because the vast majority of pixels do not contain a plume (a zero-inflated regression problem). Such problems often have the issue that the model will converge at predicting zeros everywhere. Thus, the inclusion of the mask prediction helps to prevent this. The ensuing model is composed initially of a 1×1 convolutional layer for a similar reason as its inclusion in the UNet model (see section 2.4.1). Following on from this are 2 ResNet layers (He et al., 2016), which are characterised by double-layer skip connections, ReLU activation functions, and batch normalisation (Figure 4). A ResNet architecture was selected for this portion of the model as it is known to be lightweight and powerful at regression predictions in computer vision.

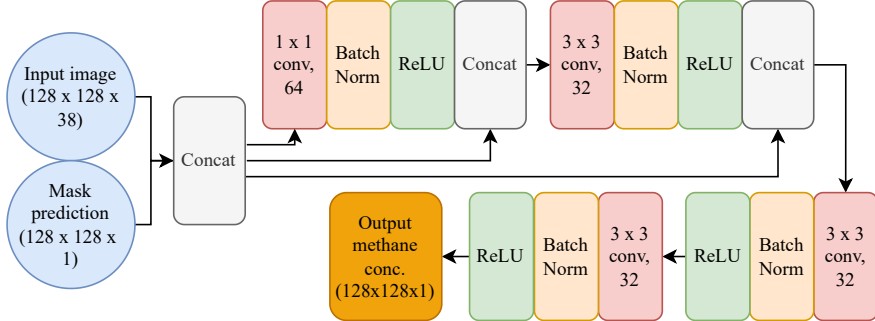

287

**Figure 4: Architecture of the deep neural network for the ResNet portion of the model. 1 × 1 conv, 64 refers to a convolutional filter with kernel size 1 × 1 and 64 filters. Batch Norm refers to a batch normalisation layer and Concat refers to a concatenation between the inputs to that layer. ReLU refers to the Rectified Linear Unit activation function.**





### 2.5.3 Estimating emission rate of plumes

The prediction of the binary classification of plume/not plume involved an architecture identical to the one presented in this section (except the final activation layer was sigmoid, not ReLU). The inputs to the emission rate portion of the model are the outputs from all the previous stages of the model concatenated with the input image. This is to ensure that more information is available to the model to accurately estimate emission rates. Following on from this is the 1×1 convolutional layer, which was included for the same reason as in the previous stages of the model (see section 2.4.1). This is followed by the decoder part of the model, in which a convolutional layer is followed by batch normalisation, ReLU activation, and max pooling, which is done 7 times with increasing filters every 2nd loop. These layers encode features about the methane plumes and reduce the dimensionality of the tensors. Finally, there is a dense layer and ReLU activation to collect all information obtained and output a single floating-point number as the predicted emission rate (Figure 5).

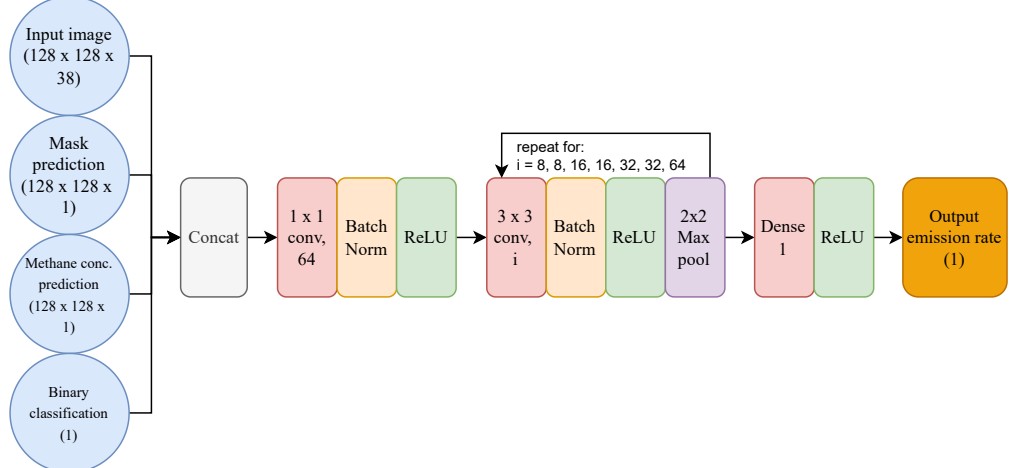

**Figure 5: Architecture of the deep neural network for the emission rate prediction of the model. 1 × 1 conv, 64 refers to a convolutional filter with kernel size 1 × 1 and 64 filters. Batch Norm refers to a batch normalisation layer, Concat refers to a concatenation between the inputs to that layer, and 2 x 2 Max pool refers to a max pooling layer with pool size 2. ReLU refers to the Rectified Linear Unit activation function.**

## 3 Results

### 3.1 Application of neural network to simulated plumes

The total training/validation dataset consisted of 9700 images, 80% of which were reserved for training and the remaining 20% for validation. After each iteration of the model through the training dataset (known as an *epoch*), the model was tested on the validation dataset. If the loss of the model when tested on the validation dataset was lower than the lowest loss



previously recorded, the weights of the model were updated. Thus, at the end of the training procedure, the best model was
saved. Each of the stages of the model depicted in Figure 2 were trained separately in descending order, where the weights of
the other stages did not vary. This was done so that the most accurate predictions were being produced from the earlier layers
so that no errors from insufficient training would propagate through the model because the outputs are concatenated with the
satellite data in later parts of the model.
Once training was complete, the model was tested on an additional 2000 images not seen during training sampled randomly
from a uniform distribution of emission rates from 500 to 10 000 kg hr$^{-1}$. 36/2000 of the images had a maximum methane
concentration under the $4\times10^{18}$ molecules cm$^{-2}$ threshold imposed during training, however they were still included in the
testing to determine if they can still be detected by the model. The model is able to accurately locate and identify the shape
of methane plumes in the test dataset (Figure 6).







**Figure 6: Example images and predictions taken from the test dataset. Images are 3840x3840m composed of 128x128-pixel tiles. True emission rates and initial wind speeds are (a) 8068 kg hr⁻¹ , 1 ms⁻¹, (b) 1484 kg hr⁻¹ , 1 ms⁻¹, (c) 7673 kg hr⁻¹ , 5 ms⁻¹, (d) 6270 kg hr⁻¹ , 4 ms⁻¹. Retrieved methane comes from the retrieval described in section 2.2. RGB image courtesy of PRISMA © (Italian Space Agency).**

The total methane column enhancement in the images was well estimated, where total estimated methane was closely correlated with the ground truth (Figure 7) with a tendency to slightly overestimate column values.



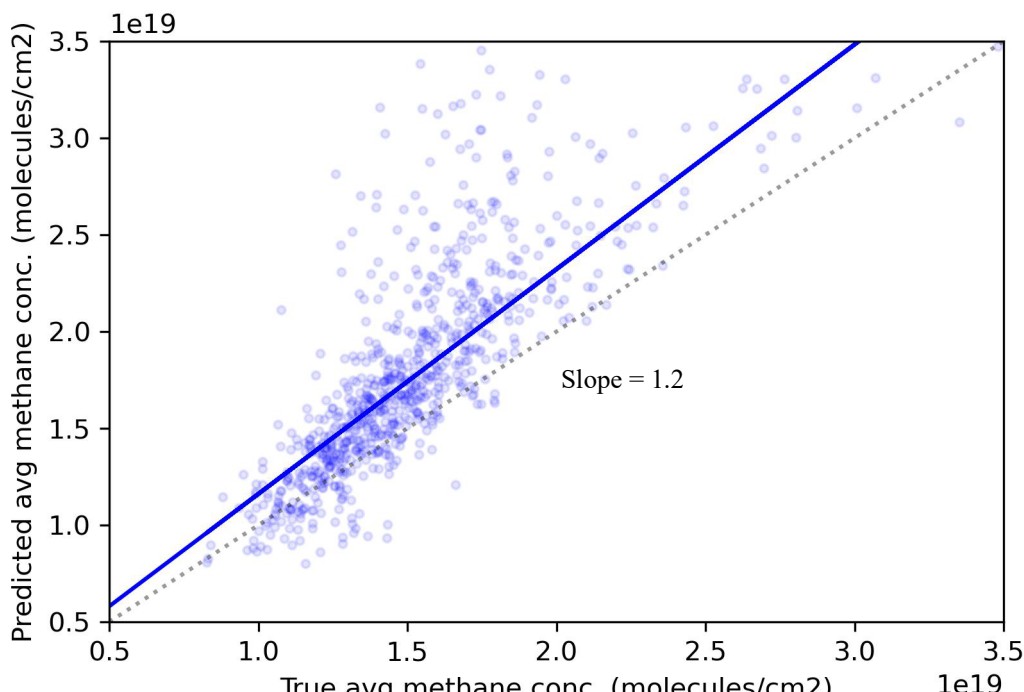

**Figure 7: Scatter Plot of mean methane concentration predicted vs true.**

In the binary classification part of the model, we assess its success using the F1-score, precision and recall, which are defined

as follows:

F1 = TP/(TP+0.5*(FP+FN)),                                                                                          (6)

Precision = TP/(TP+FN),                                                                                            (7)

Recall = TP/(TP+FP),                                                                                               (8)

In the binary classification part of the model, the F1-score, precision, and recall were 0.95, 0.96 and 0.92, respectively (Table

1). These statistics come from predictions made on the 2000 images with plumes in, as well as an additional 1533 images

with no plumes.

**Table 1: Confusion matrix of binary classification portion of the model broken down per image.**

|  | Plume present | No plume present |
|---|---|---|
| **Predicted plume** | 1846 | 51 |
| **Predicted no plume** | 154 | 1482 |





The distributions of the scene noise and methane concentrations in the cases where no plume was predicted but a plume was
present (false negative) reveal slightly lower than average scene noise and much lower than average maximum methane
concentration (Table S2). However, in the cases where a plume was predicted but no plume was present (false positive), scene
noise is not noticeably different (Table S2).

The actual vs predicted emission rate has a slope of 0.83 with a relatively small spread about the line of best fit (std = 1447
kg hr$^{-1}$). This means that there is a tendency for underestimating emissions with a mean absolute percentage error in emission
rate of 23.7% (Figure 8). This bias in the slope is possibly a result of training the model on images without plumes as well as
those containing plumes.

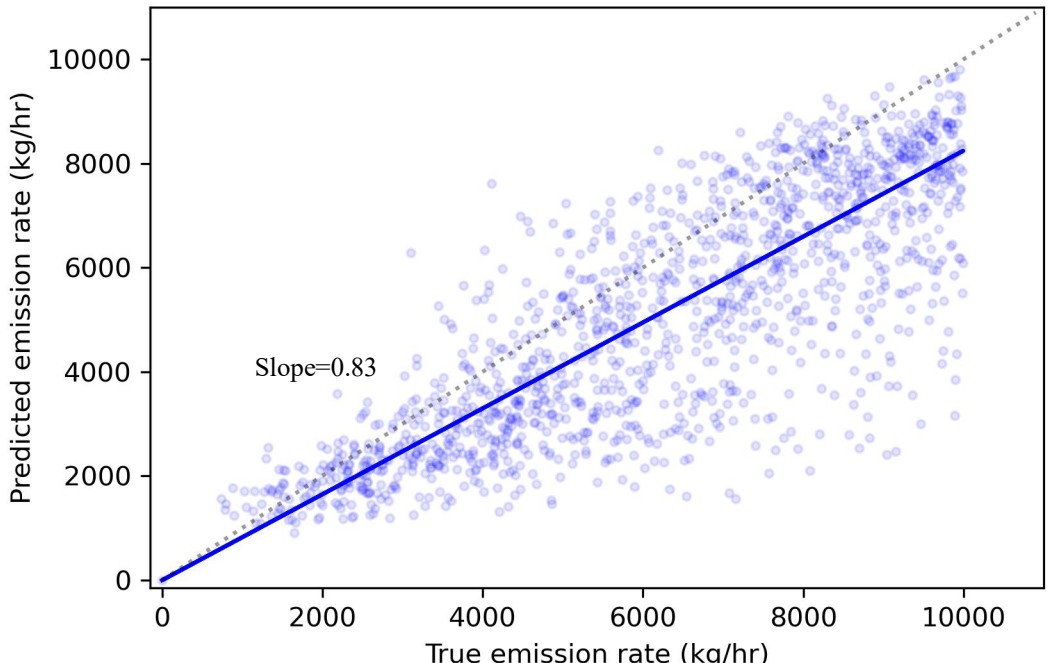


**Figure 8: Actual vs predicted emission rate using the deep learning model. Line of best fit calculated using Huber loss so outliers
do not have an inordinate influence on the slope.**
The absolute emission rate error increased in magnitude as the emission rate increased (Figure 8), as one might expect. The
percentage error was largest in magnitude for the smallest emission rates (500-999 kg hr$^{-1}$), but the distribution remained
relatively consistent above 2000 kg hr$^{-1}$, with a median error of 25% and interquartile range of 40% error (Figure 9). The error
in percentage emission rate had a positive bias for emission rates under 1000 kg hr$^{-1}$ and a negative bias for emission rates
over 2000 kg hr$^{-1}$ (Figure 9).






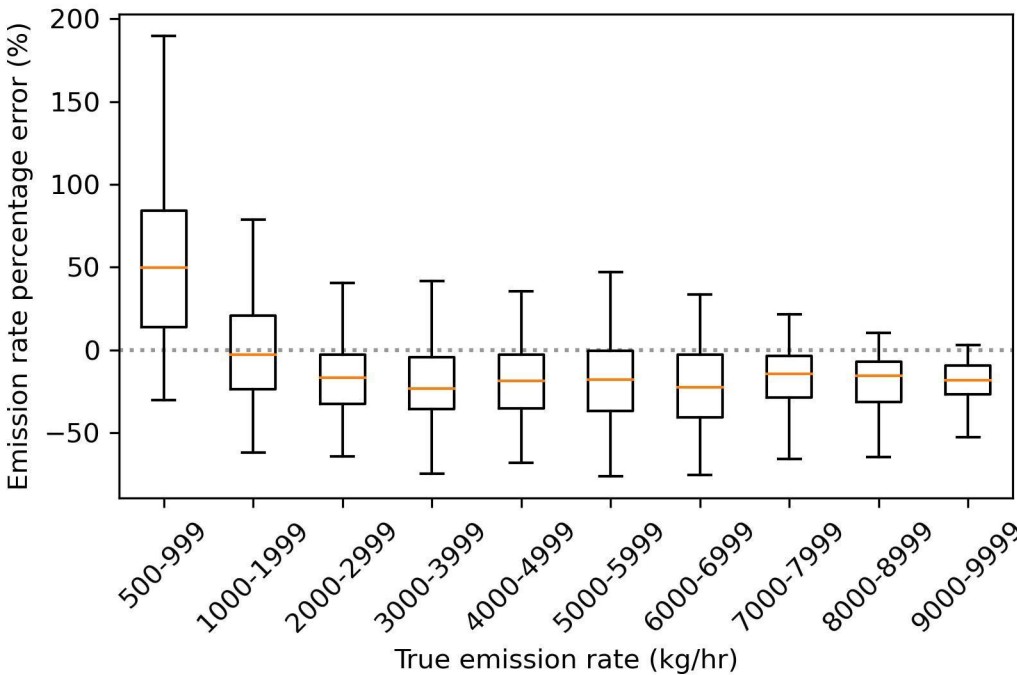


**Figure 9: Error in emission rate predictions from the deep learning model as a function of true emission rate. Positive values indicate predicted emission rates being larger than true emission rates. Top panel shows absolute emission rate error and bottom panel shows percentage emission rate error.**

**3.2 Application to real-world images**

**3.2 Application to real-world images**

The model was then tested on 40 PRISMA scenes obtained during 2020-2022 in the Korpeje oil field, Turkmenistan (37.9˚N, 53.2˚E - 39.4˚N, 55.2˚E), a well-studied area with frequent methane point source emissions plumes (Irakulis-Loitxate et al., 2022). The images were normalised in the same way that the training, test, and validation images were. 21 plumes were identified from 15 different scenes with predicted emission rates ranging from 1112-7615 kg hr$^{-1}$ (Figure 10; Table S3).




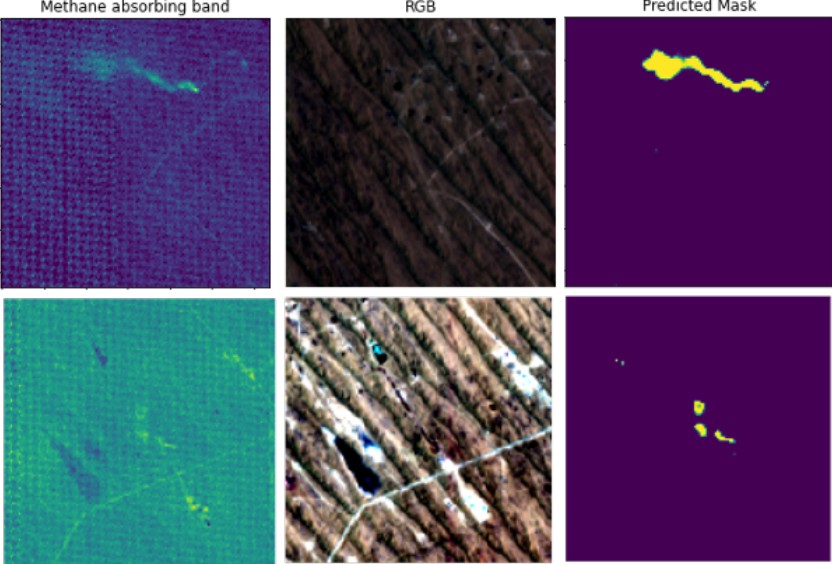

**Figure 10: Images of plumes detected by the neural network in the Korpeje oil field, Turkmenistan. Left panels depict methane retrievals, middle panels depict the RGB of the image, and the right panel depicts the mask prediction by the neural network. The predicted emission rates are (top) 7615 and (bottom) 2370 kg hr$^{-1}$. RGB image courtesy of PRISMA © (Italian Space Agency).**

Methane plume detection capability using the neural network was compared with using clustering and thresholding techniques (see section 2.2). Out of the 21 plumes, 14 were found using this approach. The neural network model took roughly 1 minute to make predictions of plume masks, methane concentrations, and emission rates of located plumes in an image of 1000x1000 pixels (900km$^2$ area) without the need for time-consuming human inspection typically needed for classical clustering approaches.

## 4 Discussion

Identification and reduction of methane emissions can have a considerable influence over the Earth's surface radiation budget and hence our efforts to mitigate climate change. Methods utilising classical approaches have had some success in detecting fossil fuel methane point sources and estimating their emissions, but the errors are high (roughly 50% error for emission rate predictions) if no accurate local wind speed information is available and often time-consuming human judgement is necessary to separate plumes from surface effects. Within the pseudo-observation dataset produced in this study, only one quarter of the images were deemed suitable to be analysed via clustering algorithms, which demonstrates its limitation for detecting methane point source emissions. In comparison, only 7.7% of the pseudo-observations were undetected by the neural network (Table 1). The neural network presented in this study was able to accurately locate simulated methane point source plumes





with a precision and recall of 0.96 and 0.92, respectively. The estimates of emission rate did not require wind speed
information, which is a major source for uncertainty in emission estimates in conventional approaches such as the IME
method, and had an average error of 23.7%, which is considerably lower than that obtained from classical methods. The
emission rate prediction error could possibly be further reduced with training on a larger dataset.

The approach used in this study differs from the approach by Jongaramrungruang et al. (2022), who directly predicted the
emission rate from the satellite data without first estimating the plume mask. However, we found that excluding these stages
dramatically worsened the model prediction, where the error in emission rate was greater than 50%. The model architecture
presented here utilises the maximum amount of information available from the training data. Possible explanations for why
the model from Jongaramrungruang et al. (2022) was nevertheless successful could include the large training data volume
available in their study (in the order of hundreds of thousands of images), which is an order of magnitude larger than that
available in this study. This larger training volume may have enabled the neural network to make the link between plume
shapes and emission rates. In addition, the spectral and spatial resolution of the aircraft imagery used in their study (AVIRIS-
NG) is substantially higher than that of PRISMA. Finally, the input bands for this study totalled 38, whereas in the study of
(Jongaramrungruang et al., 2022), only 1 band was sufficient due to the low noise in the signal in the AVIRIS-NG data and
high methane absorption in that band. Thus, it may have been easier for their neural network to learn features in the image
due to lower noise present.

When producing the training data labels for plume masks, a constant threshold was chosen for what methane concentration
constitutes a plume. However, the minimum methane concentration that is detectable likely varies depending on scene noise
and brightness. Thus, more work is necessary to quantify the most appropriate threshold. However, precise estimates of the
edges of a plume are of lesser importance than the initial identification of a plume and its corresponding emission rate.

There is a noticeable bias present in the emission rate prediction errors (Figure 8; Figure 9) which was also evident in the
study by Jongaramrungruang et al. (2022). This bias should be rectified, and future work is needed in fine tuning the neural
network training procedure to do so. Such adjustments could include modifying the emission rate loss function or the model
architecture. The model was trained only on images with a single methane point source; thus, the model is not able to
discriminate between emissions from different sources within a single 128x128-pixel image. The solution to this would be to
add in training data with multiple sources and solve the instance segmentation problem using an appropriate architecture,
such as Mask-RCNN (He et al., 2020). It is likely that the errors would be larger in general when using this approach owing
to the increased noise present.



## 5 Conclusions

In this study, we present a novel deep neural network model for identifying and quantifying methane point source emissions from PRISMA satellite data. PRISMA data has sufficient spectral and spatial resolution to identify methane plumes, while still having considerable spatial coverage and is still in operation today. These factors make PRISMA an ideal tool for methane detection and the deep neural network developed here has great potential to impact climate mitigation efforts. The model proved to be more successful with both identification and quantification than previous efforts using classical approaches. Rapid identification and quantification of methane point sources is a vital contribution to climate change mitigation, and the approach outlined here opens the door to the capability to automate methane plume detection. Our model was able to produce predictions on an area of 900 km$^2$ over real PRISMA images in less than a minute. Such a capability would vastly reduce the time and costs associated with reducing anthropogenic methane emissions.

## Acknowledgements

We acknowledge funding from the Natural Environmental Research Council (NERC). Peter Joyce, Cristina Ruiz Villena, Alex Webb, Chris Wilson, Martyn P. Chipperfield, Yahui Huang, and Hartmut Boesch are funded via the UK National Centre for Earth Observation (NE/R016518/1 and NE/N018079/1). Part of this work was carried out at the Jet Propulsion Laboratory, California Institute of Technology, under a contract with the National Aeronautics and Space Administration (NASA). Project carried out using ORIGINAL PRISMA Products - © Italian Space Agency (ASI); the Products have been delivered under an ASI Licence to Use.

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
