# Peer review of "and quantify emissions from PRISMA hyperspectral satellite"

_EGUsphere, 2022_

## Author Comment (AC1)

Dear reviewers,

Thank you for your helpful comments. Below we list our responses to each comment. **Please note that the line numbers given refer to the lines in the Tracked_changes.pdf file.** This was done to prevent further confusion regarding line numbers (I think different versions of MS word deal with them differently).

**L. 21: 1 km2 seems quite large for point sources except landfills. Point sources are usually on the order of m2. Are you referring to the area of a detected plume, or just trying to include a range of source types?**
Added clarification (L24)

**L. 23: Classical methods are certainly time-consuming, but are they inaccurate? I don't think so, e.g., see Sherwin et al. (2022): https://eartharxiv.org/repository/view/3465/**

This paper has now been published (https://www.nature.com/articles/s41598-023-30761-2). We have cited it (L90) and revised statement (L26)

L. 46: One of Daniel Cusworth's papers would be an appropriate reference for the temporal emission variability of point sources. E.g., consider Cusworth et al. (2021): https://pubs.acs.org/doi/10.1021/acs.estlett.1c00173
Added (L52)

L. 67: "prone to errors owing to the substantial human intervention required". This strikes me as a bit backwards; I would expect human intervention to produce the highest quality plume detection/delineation, just as human labeling of (for example) photos of cats and dogs produces the most accurate results. The problem is that human intervention is costly.
Good point, errors are lower and less frequent when scenes are analysed by experts, thus we changed to 'can be prone to errors without sufficient training' (L77)

L. 73-74: Varon et al. (2018) reported 15-65% additional error from uncertain wind speed.
Added in (L85)

**L. 198-201: If I understand correctly, the conversion of methane concentration to change in radiance does not account for the plume vertical distribution – i.e., the plume is first vertically integrated and then a single pressure/temperature value is**

**used for the radiative transfer calculation. Do you expect this simplification to have a negligible effect?**

This is correct. This is a simplification that we make here which could be improved in future studies. However, we expect that this will only introduce minor differences as the plume vertical extend will typically kept within the boundary layer.

Added a statement (L233)

L. 219-226: I found this section hard to understand. Can you explain more clearly how a classical threshold helps with / is applied in the training procedure? Is it simply to create the ground truth plume masks for the automated plume masking task?
Correct, we have added in clarification (L254)

L. 253-255: Suggest identifying which CNNs are encoder CNNs earlier, because it wasn't clear to me that you are referring to the CNN for binary detection and the emission rate estimator (if I'm not mistaken). And the U-Net also involves an encoder branch, so this feels a bit ambiguous.
Added in clarification [290-292]

Fig. 2 & Fig. 5: how does the logical output of the binary plume detection model get appended to the data cube that is passed to the emission rate estimator? Is it just a uniform channel of 1's or 0's?
Added in clarification (L340-342)

Section 2.5.1: It's not clear to me what purpose this 1x1 convolution serves, can you rephrase?
Added clarification (L306-307)

L. 297: Shouldn't this be the "encoder" part of the model, not decoder? It's encoding the input data to smaller dimensionality before applying the dense layer.
Corrected (L345)

**L. 333: Can you provide some discussion of why the model might tend to overestimate concentration? Could be 1-2 sentences.**
Added some discussion (L385; L412-414)

Table 1 & general: I found myself wondering at several points whether "binary classification" refers only to the yes/no plume detection test, and/or to the U-net binary segmentation task (binary classification per pixel). Please clarify this distinction.

Added clarification in the table caption and elsewhere in the manuscript (adding 'whole image') (L266; L279; L291; L293; L336; L340; L393; L399; L403)

**Fig. 8: It's interesting that your network overestimates concentrations but underestimates emissions. If the no-plume images are really to blame for this, then they seem to have a strong effect. This could be checked by retraining the final network without no-plume images. Whether or not you do that, I feel this finding deserves a bit more discussion.**

Added more discussion on the methane concentration overprediction and emission rate underprediction as previously discussed (L385; L412-414). Important to note that while the output of the methane concentration model does feed into the emission rate, the plume mask prediction and satellite image were generally more influential on the emission rate predictions (possibly because the error in the methane concentration was higher than the mask).

Fig. 10: Are the retrieval fields in the left column of the figure from your network or a physics-based retrieval?
Added clarification (L441)

L. 381: Are you saying that your network found 14/21 plumes? Or was that the result of a classical detection scheme?
Added clarification (L447)

**L. 390-391: "only one quarter of the images were deemed suitable to be analysed via clustering algorithms". On what basis?**

This is from the visual inspection. The presence of interference effects from surface features make plume detection tricky. Added this detail to the text (L458)

**L. 391 & elsewhere: "clustering algorithms". It's not clear to me what you mean by this. Classical thresholding is clear, but what clustering techniques are you referring to?**

It is to separate different plume-like features that are obtained with the PCA retrieval method due to surface interference.

Added clarification for using DBSCAN clustering (L140; L446)

L. 393: "accurately locate" I thought those statistics were for the binary detection, but this wording would suggest they were for the U-Net segmentation. See my comment above on "Table 1 & general". Can you clarify?
Added clarification (L460)

**L. 396: "which is considerably lower than that obtained from classical methods". Is that true? 25% mean absolute error seems similar to what's reported in the Sherwin et al. (2022) controlled release study, for which participants used classical methods. Furthermore, the classical methods appear to have near-zero bias (Varon et al., 2018; Sherwin et al., 2022), despite significant error spread, whereas your method has 17% bias. And your 40% interquartile range also seems not so different from a ~50% error standard deviation.**

Varon et al., 2018 is for the GHGsat instrument so that it can not be compared directly. They state an error range of 15 %–50 % in the IME method and 30 %–65 % in the cross-sectional flux method. The main cause for these errors is wind knowledge.
They also did only an ideal case without any surface features.

Sherwin: The release experiment is for a bright site which gives favourable results. The quoted error for PRISMA has a mean error of 27% with a min of -20% and max. of 110%. We have modified the statement slightly (L472) to be more specific that we are only talking about our own classical method. We could include the information given above but it is hard to make direct comparisons when the choices of scene and satellites used are different.

Is the low bias for high emissions and high bias for low emissions also a question of training?

Discussed earlier in this document, added comments (L385; L412).

**Adding to my previous comment: How does the error standard deviation (spread) of your method compare with the ~50% classical error? Or put the other way, how does your 25% mean absolute relative error compare to the same quantity for the classical methods? My impression is that your ML network is much more efficient than classical methods, but likely less accurate -- and it's not clear to me that it's more precise (except in application to multiple nearby point sources, where, as Jongaramrungruang**

**et al. 2019 point out, using a wind-direction-independent method leads to error cancellation in the emission sum). A more careful comparison of the merits of this work compared to previous methods would be helpful.**

The 50% error quoted for the classical approach can be directly compared with the 25% error for the neural network because both are mean absolute errors. Added in this clarification (L455).

**L. 399-410: "where the error in emission rate was greater than 50%". But as you say, Jongaramrungruang et al. did much better than that. In addition to the possible reasons you give, could it be because they did plume detection/quantification on methane retrieval fields, whereas your model mainly relies on multi-channel radiance data? I.e., they combined physics and ML methods by processing the methane retrieval before applying their network, whereas your network is fully machine-learned. That would be an interesting finding, if true.**

This is for AVIRIS-NG which has much better instrument parameters for CH4 detection that PRISMA (e.g. higher spectral resolution)

**General comment #2: Just a perspective to consider: I feel that the primary strength of your methodology is its potential for rapid application to increasingly large satellite datasets for methane, rather than achieving the most accurate and precise point source masks and emission rate estimates. I would expect careful human analysis to be generally superior in that respect (accuracy + precision), but perhaps not by much, and certainly highly inefficient compared to your work; clearly there aren't enough human analysts to carefully process all the data from PRISMA, EMIT, EnMAP, Sentinel-2, Landsat, etc. That your method automates plume detection/quantification with performance comparable to human analysis, even with some low bias, is a major accomplishment.**

Reduced claims of accuracy in the text and reframed the problem (L77; L91; L501 ;L509)

**L. 430: Again, it's not clear to me that your method is more successful than classical approaches in quantifying emission rates, given the combination of low bias and prediction spread you find. If I'm wrong, please just clarify the comparisons throughout the manuscript.**

Reduced claims of accuracy in the text as mentioned above

L. 284: section 2.4.1 --> section 2.5.1
Corrected (L299)

L. 296: section 2.4.2 --> section 2.5.2
Corrected (L321)

L. 369: section heading duplicated
Corrected (L431)

**References**
**Cusworth et al. (2021) https://pubs.acs.org/doi/10.1021/acs.estlett.1c00173**
**Jongaramrungruang et al. (2019) https://amt.copernicus.org/articles/12/6667/2019/**
**Sherwin et al. (2022) https://eartharxiv.org/repository/view/3465/**

**Testing on real data: I understand that the proposed methodology is expected to be globally applicable. PRISMA scenes from a wide range of site conditions are actually used for algorithm training (Table S1). However, only results from real plumes in Turkmenistan are presented. Turkmenistan is considered an optimal study region for plume detection, since surfaces are typically bright and homogeneous, and plumes are large. For the readers to get a better impression of the method's performance, it would be great to see how the it works in other sites. In particular, the authors could use the PRISMA scenes and plume detections in the Permian Basin and the Shanxi region reported by Guanter et al. (2021), to which we gave access to the authors. Why were results from those sites not included?**

We include in a separate document the results from these sites. The neural network was able to find most of the plumes from Guanter et al (2021), but not all. The neural network did also find some strong candidates for plumes that were not given in Guanter et al (2021).

It is worth pointing out that the large plume identified in Guanter et al (2021) in Turkmenistan is in an area with high cloud cover and our clustering algorithm was also unsuccessful in finding the plume. Future research direction could be to train a neural network with cloudy scenes to improve accuracy in such cases. One additional improvement could be to train the neural network on a larger volume of scenes to improve performance in different landscapes. Added in this detail (L489).

The neural net appears to perform as well as, or better than our PCA approach in some situations, but worse in others. As pointed out by reviewer #1, the key advantage of this work is that it speeds up the detection process as it's not possible for all PRISMA scenes to be manually examined by experts.

**Overall presentation of results: I feel that a stronger effort could be done in the analysis and presentation of results from real data. For example, by providing more information on the comparison between the proposed AI-based method and that of the existing "clustering and thresholding" methods. One could show the potential and limitations of each method, or where the AI-based method does not outperform the supervised method. Also, it would be useful to see more concentration enhancement maps, especially for the plumes which where not detected by the method. Detecting 14 out of 21 plumes with flux rates >1 t/h in Turkmenistan doesn't sound that impressive, and it would be good if this could be discussed further.**

The focus of the paper is on the method and simulations. The purpose of the application to real data was included as a demonstration and not a full analysis. A general application of the neural network to a large range of real scene might require a larger number of training scenes as already mentioned above.

Although our PCA method works well, further improvements are possible by adopting a cluster-based PCA approach which could lead to a larger plume detection rate.

**L21: "order of km2" the sources or the plumes? I guess the latter?**
Dealt with (L24)

**L29: what is a F1-score?**
Defined in equ 6 (L395), no space in abstract to define it

**L20: (line numbering restarted in p4): regarding PRISMA CO2 retrievals, this https://agupubs.onlinelibrary.wiley.com/doi/full/10.1029/2020AV000350 could be cited**

Added in citation (L70)

**L45: Section 2.2: could you introduce here what this retrieval step is needed for? Not clear to me until much later in the document**
Added in a line to introduce and frame it (L139)

**L57: among data-driven retrieval methods, I think matched-filter retrievals are used in more studies so far than the PCA-based retrievals. Any reason why you chose the latter?**

It is true that different methods exist. We believe that the PCA method is a methodologically good solution that is also simple to implement. Further improvements of this PCA based method is possible by adopting a cluster-based PCA approach. We are not aware of strong indications that the matched-filter retrieval is superior.

**L74 (p7): I think the per-column processing is actually more important because of striping (column-wise changes in the instrument's radiometric response)**
Added into text (L201)

L74 (p11, line numbering restarted again): There is no Sec 2.4.1.
Fixed

**Fig. 6: 3 of the plumes / emission rates selected for this figure are actually huge. Consider to use more normal cases of 1-3 t/h?**
**The authors might like to discuss this recent preprint on the same topic**
https://res.cloudinary.com/diywkbi34/image/upload/v1669115401/Marketing/COP27/Kayrros%20Science_%20Detecting%20Methane%20Plumes%20using%20PRISMA:%20Deep%20Learning%20Model%20and%20Data%20Augmentation.pdf
Due to the limitations of PRISMA imagery, smaller plumes are harder to detect and less visually interesting considering the spatial resolution. The pre-print provided is very short and no results shown, so maybe not appropriate to discuss in the paper.

NOTE:
We also corrected the y-axis of Figure 1 from Altitude to Vertical distance.